# An Integrated Model including the ROX Index to Predict the Success of High-Flow Nasal Cannula Use after Planned Extubation: A Retrospective Observational Cohort Study

**DOI:** 10.3390/jcm10163513

**Published:** 2021-08-10

**Authors:** Young Seok Lee, Sung Won Chang, Jae Kyeom Sim, Sua Kim, Je Hyeong Kim

**Affiliations:** 1Department of Internal Medicine, Division of Pulmonary, Allergy, and Critical Care Medicine, Korea University Guro Hospital, Seoul 08308, Korea; avonlea76@korea.ac.kr (Y.S.L.); kyoum-2@hanmail.net (J.K.S.); 2Artillery Brigade, The 5th Infantry Division, 6 Corps, Ground Operation Command, Republic of Korea Army, Yeoncheon 11132, Korea; lego42@naver.com; 3Department of Critical Care Medicine, Korea University Ansan Hospital, Ansan 15355, Korea; sua0047@gmail.com

**Keywords:** high flow nasal cannula, extubation, hypoxemia, ROX index, reintubation, oxygen therapy

## Abstract

High-flow nasal cannula (HFNC) therapy is commonly used to prevent reintubation after planned extubation. In clinical practice, there are no appropriate tools to evaluate whether HFNC therapy was successful or failed after planned extubation. In this retrospective observational study, we investigated whether the use of the ROX index was appropriate to differentiate between HFNC success and failure within 72 h after extubation and to develop an integrated model including the ROX index to improve the prediction of HFNC success in patients receiving HFNC therapy after planned extubation. Of 276 patients, 50 patients (18.1%) were reintubated within 72 h of extubation. ROX index values of >8.7 at 2 h, >8.7 at 6 h, and >10.4 at 12 h after HFNC therapy were all meaningful predictors of HFNC success in extubated patients. In addition, the integrated model including the ROX index had a better predictive capability for HFNC success than the ROX index alone. In conclusion, the ROX index at 2, 6, and 12 h could be applied to extubated patients to predict HFNC success after planned extubation. To improve its predictive power, we should also consider an integrated model consisting of the ROX index, sex, body mass index, and the total duration of ventilator care.

## 1. Introduction

The use of a high-flow nasal cannula (HFNC) is useful for oxygen delivery in critically ill patients. In clinical practice, HFNCs are used to manage acute respiratory failure (ARF), prevent hypoxemia during procedures, and prevent postextubation hypoxemia [1,2,3,4,5,6,7]. As almost 20% of patients who undergo planned extubation require reintubation because of hypoxemia, and the HFNC is a useful oxygen delivery modality in this setting, HFNCs are commonly used to prevent reintubation after planned extubation [4,5,8,9,10,11,12]. As the HFNC can maintain the oxygen concentration despite a patient’s declining status, clinicians may be hesitant to reintubate even if the patient’s condition deteriorates. Therefore, reintubation could be delayed in patients with HFNC application, and it could be associated with increased mortality [13,14,15]. Therefore, intensivists should carefully evaluate whether patients will need to be reintubated after the application of HFNC to prevent a delay in reintubation. However, there are no appropriate tools to evaluate whether HFNC therapy was successful or failed after planned extubation in actual clinical practice.

The ROX index (Spo_2_/Fio_2_/respiratory rate) has recently been developed and validated to predict whether HFNC therapy will be successful within 24 h of commencement in patients with pneumonia-related ARF [16,17]. As the ROX index has been validated only in patients with pneumonia-related ARF, it cannot be applied to other clinical settings. As the ROX index is an effective bedside tool for predicting HFNC success, it should be validated for use in various clinical settings. We hypothesized that the ROX index would be useful to predict the success of HFNC in extubated patients and that a model composed of the ROX index and other factors would improve the predictive power for HFNC success in all extubated patients. This study investigated whether the use of the ROX index was appropriate to differentiate between HFNC success and failure within 72 h after extubation, and it aimed to develop an integrated model including the ROX index to improve the prediction of HFNC success in patients receiving HFNC after planned extubation.

## 2. Materials and Methods

### 2.1. Study Overview

This was a retrospective observational cohort study of patients who were immediately treated with HFNC after planned extubation in the medical intensive care unit (ICU) at Korea Medical Center, a 1075-bed tertiary referral hospital in Seoul, Republic of Korea, between January 2017 and December 2019. This hospital has a 20-bed semi-closed medical ICU with an annual ICU admission rate of 400 patients and an annual ventilator application rate of 300 patients. The ratio of patients to nurses is approximately 2.5:1, and three intensivists oversee nearly all patients in the medical ICU. All patients in this study were progressively weaned from mechanical ventilation at the discretion of the intensivist based on the weaning protocol of the hospital. Intensivists evaluated the need for reintubation on the basis of the reintubation protocol in patients showing clinical deterioration after planned extubation.

### 2.2. Ethics Statement

This study was performed in strict accordance with the principles expressed in the Declaration of Helsinki. This study was approved by the Institutional Review Board of Korea Medical Center (approval no.: 2020GR0518, 4 November 2020). We ensured the protection of patient privacy and anonymity. The need for informed consent was waived because of the retrospective nature of the study.

### 2.3. Patients

The inclusion criteria were as follows: age ≥ 20 years; patients who were treated with HFNC after planned extubation; patients receiving mechanical ventilation in the medical ICU; and patients with ROX index measurements recorded at 2, 6, 12, and 24 h. The exclusion criteria were as follows: patients who were not extubated because of death or because they were transferred to another hospital while intubated; patients who were treated with low-flow oxygen therapy (e.g., nasal cannula or mask) or noninvasive ventilation (NIV) after extubation; and patients lacking electronic medical records (EMRs) that were needed to calculate the ROX index. Patients were followed up until death or hospital discharge.

### 2.4. Definitions and Description of Variables

The ROX index was defined as the ratio of Spo_2_/Fio_2_ to the respiratory rate (breaths/min). The ROX index was measured at 2, 6, 12, and 24 h after commencement of HFNC therapy [17]. Body mass index (BMI) was calculated by dividing body weight in kilograms by the height in meters squared. Comorbidities were defined as a previous diagnosis by doctors, and hospital mortality was defined as death during the interval from admission to discharge. HFNC success was defined as the continuation of HFNC with a stable condition within 72 h after extubation or successful discontinuation of HFNC because of stable condition. HFNC failure was defined as a transition to mechanical ventilation with reintubation within 72 h after extubation.

### 2.5. Weaning Protocol

Patients ventilated for more than 24 h underwent a spontaneous awakening trial (SAT). Patients who successfully passed the SAT were evaluated for the possibility of weaning using the weaning criteria, i.e., the resolution or improvement of the condition leading to intubation, hemodynamic stability (systolic blood pressure between 90 and 160 mmHg and heart rate < 140/min without vasopressors or with low doses of vasopressors), respiratory stability (oxygen saturation > 90% with Fio_2_ ≤ 0.4, respiratory rate < 35/min, and spontaneous tidal volume > 6 mL/kg) [18,19,20]. Patients who fulfilled the weaning criteria underwent a 30-min spontaneous breathing trial (SBT) with positive support ventilation of 8–10 cm H_2_O and a positive end-expiratory pressure of 6 cm H_2_O. Patients who successfully completed the SBT were extubated. If the patient did not tolerate the SBT, we changed the ventilator to control mode. The criteria for failure to tolerate the SBT were agitation, anxiety, mental status change, respiratory rate > 35/min and/or the use of accessory muscles, oxygen saturation by pulse oximetry < 90% with Fio_2_ > 0.4, or the development of arrhythmia [18,19,20].

### 2.6. Device Settings and Reintubation Criteria

High flow was provided by Airvo 2 (Fisher and Paykel Healthcare, Laval, QC, Canada). In extubated patients, HFNC therapy was initiated at a flow rate of 50 L/min with a Fio_2_ of 0.5. After the commencement of HFNC therapy, the Fio_2_ was titrated by targeting Spo_2_ > 93%. The flow rate was adjusted according to the patient’s clinical condition. Starting 30 min after commencement of HFNC therapy, the intensivist evaluated the patient for the need for reintubation for 72 h based on the patient’s clinical condition. Reintubation was considered in patients with a decreased level of consciousness, cardiac arrest/malignant arrhythmia, severe hemodynamic instability, inability to clear secretions because of respiratory fatigue, and/or worsening respiratory condition (Pao2 < 60 mmHg, pH < 7.3, or respiratory rate > 35 breaths/min, despite an HFNC flow rate of >50 L and a Fio_2_ > 0.7).

### 2.7. Statistical Analysis

The descriptive statistics are presented as medians (25th and 75th percentiles) or numbers (percentages). The Fisher’s exact test was used to analyze categorical data, and the Mann–Whitney U-test was used to compare continuous data. Receiver operating characteristic (ROC) curves were used to assess the accuracy of different variables for the classification of HFNC success. The cut-off levels of the ROX index and other factors in the integrated model were calculated using the Youden index of ROC analysis to determine the optimal diagnostic accuracy for these factors. Kaplan–Meier curves with log-rank tests were used to determine the probability of reintubation in patients with a different ROX index according to the cut-off level for optimal diagnostic accuracy (regarding the differentiation between HFNC success and failure). Cox proportional hazard analysis using backward elimination was used to identify independent factors of HFNC success. Independent variables and those with *p*-values < 0.05 in the univariate analyses were included in the multivariate analyses. The results are summarized as adjusted hazard ratios (HRs) and 95% confidence intervals (CIs). We investigated the best model that included other factors that could be used to accurately predict HFNC success after the commencement of HFNC therapy in extubated patients. The discriminatory power of each model was assessed using Harrell’s C-index. The areas under the curve (AUCs) and results were evaluated and compared using the bootstrap method [21,22]. In two-tailed tests, *p* < 0.05 was taken to indicate significance. All statistical analyses were performed using SAS 9.4 (SAS Institute, Cary, NC, USA).

## 3. Results

### 3.1. Clinical Characteristics of the Study

During the study period, 880 patients were treated with mechanical ventilation in the medical ICU. Of these patients, 222 were excluded because they died prior to extubation or were transferred to another hospital while intubated, 52 were excluded because they lacked EMRs that were needed to calculate the ROX index, 288 were excluded because they were treated with low-flow oxygen therapy after extubation, and 42 were excluded because they were treated with NIV after extubation. Therefore, 276 patients were ultimately included in the study (Figure 1).

The clinical characteristics of the 276 patients are listed in Table 1. Fifty of these patients (18.1%) were reintubated within 72 h of extubation. Patients in the HFNC success group had a higher BMI and were more likely to be female than those in the failure group. The severity-related variables (e.g., age, APACHE II score at admission, and Charlson comorbidity index) were similar in both groups. At the time of extubation, the vital signs, SOFA score, Glasgow Coma Scale, analgesia use, vasopressor use, and hemodialysis use were similar in both groups. Most patients in the HFNC success group were recovering from acute respiratory distress syndrome based on the Pao_2_/Fio_2_ ratio at the time of extubation and had a shorter total duration of ventilator care compared with patients in the HFNC failure group. In addition, patients in the HFNC success group were treated with HFNC for longer than those in the failure group. Most patients in the HFNC failure group were reintubated within 24 h of extubation.

The clinical outcomes are shown in Appendix A. In this study, 29 patients (10.5%) died in the ICU with a hospital mortality rate of 27.9%. Both the ICU mortality and hospital mortality rates were significantly higher in the HFNC failure group than in the HFNC success group. In addition, patients in the HFNC success group had a shorter ICU stay than those in the HFNC failure group.

### 3.2. Respiratory Variables for Predicting the Success of HFNC Therapy within 72 h

After 2, 6, 12, and 24 h, 270 (97.8%), 243 (88%), 233 (84.4%), and 156 (56.5%) patients were still on HFNC, respectively. As the number of patients with HFNC at 24 h corresponded to almost half of the total study population, we excluded the ROX index at 24 h from the analysis.

With regard to several respiratory variables, there were significant differences between the HFNC success and the HFNC failure groups in Spo_2_/Fio_2_ ratio, respiratory rate, HFNC settings (e.g., flow and Fio_2_), and the ROX index. Patients in the HFNC success group had a higher Spo_2_/Fio_2_ ratio and ROX index but a lower respiratory rate than those in the HFNC failure group after the commencement of HFNC therapy (Appendix A).

We analyzed the levels of respiratory variables using the ROC curve to evaluate the accuracy of the differentiation between HFNC success and failure. The AUC in the ROC curve analysis indicated that the ROX index was a more accurate predictor of success than the other respiratory variables (AUC > 0.7). In particular, the ROX index at 12 h showed greater accuracy in predicting HFNC success or failure than all ROX index measurements (AUC = 0.729) (Table 2)**.**

The optimal ROX index cut-off value at 2 and 6 h was 8.7, while that at 12 h was 10.4. The performance of the optimal cut-off values at different time points is shown in Appendix A. At 12 h after the commencement of HFNC therapy, the sensitivity, specificity, positive predictive value, and negative predictive value of an ROX index >10.4 were 55.2%, 81.3%, 94.9%, and 22.4%, respectively.

### 3.3. Factors Predicting Reintubation within 72 h of Commencement of HFNC Therapy

The Kaplan–Meier curves for the cumulative risk for reintubation according to the cut-off ROX index values at each time point showed that patients below the cut-off value were more likely to require reintubation than those above the cut-off value (log-rank test, *p* < 0.001; Figure 2).

In addition, we investigated the factors associated with predicting reintubation within 72 h of the commencement of HFNC therapy using a Cox proportional hazard model. In the univariate analysis, male sex, low BMI, and low Pao_2_/Fio_2_ ratio at the start of HFNC, as well as a long total duration of ventilator care were meaningful factors for predicting reintubation within 72 h (Appendix A). The ROX indices at 2, 6, and 12 h were all meaningful factors for predicting reintubation within 72 h. The ROX index at each time point was also predictive of reintubation within 72 h after adjusting for other meaningful factors determined in the univariate analysis (Table 3).

In the multivariate analysis, low ROX index, male sex, low BMI, and long total duration of ventilator care were meaningful factors for predicting reintubation within 72 h (Appendix A).

### 3.4. Prognostic Capabilities of Models Using the ROX Index and Potential Factors for Predicting HFNC Success

We used Harrell’s C-index to compare the efficacy of predicting HFNC success using the ROX index alone versus other meaningful factors (e.g., the total duration of ventilator care, female sex, and BMI) combined with the ROX index. Models 1–4 included the ROX index at 2 h and other factors, models 5–8 included the ROX index at 6 h and other factors, and models 9–12 included the ROX index at 12 h and other factors. The combination of the ROX index and other factors had a better capability of predicting HFNC success than the ROX index alone. The differences in the AUCs for the models including other factors compared with the ROX index alone model were statistically significant (Table 4).

The optimal cut-off values for BMI and the total duration of ventilator care were 21.53 kg/m^2^ and 8 days, respectively. Figure 3 shows the changes in the predictive value for HFNC success within 72 h according to the number of factors associated with HFNC success at each time point. Figure 3a shows the changes in predicted values according to changes in factors, such as an ROX index at 2 h of >8.7, female sex, BMI > 21.53 kg/m^2^, and total duration of ventilator management ≤ 8 days. Figure 3b shows the changes in predicted values according to changes in factors, such as an ROX index at 6 h of >8.7, female sex, BMI >21.53 kg/m^2^, and a total duration of ventilator management of ≤8 days. Figure 3c shows the changes in predicted values according to changes in factors, such as an ROX index at 12 h of >8.7, female sex, BMI > 21.53 kg/m^2^, and a total duration of ventilator management if ≤8 days. Sequential increases in factors associated with the success of HFNC (e.g., the ROX index at 2 h, 6 h, and 12 h, female sex, BMI > 21.53 kg/m^2^, and a total duration of ventilator care of ≤8 days) improved the predictive capability for HFNC success within 72 h. The integrated model including all factors showed good predictive ability, reaching almost 100%.

## 4. Discussion

This study focused on the clinical value of the ROX index to predict the success of HFNC therapy within 72 h of commencement in extubated patients, and the development of an integrated model including the ROX index to improve the predictive power for HFNC success. An ROX index of >8.7 at 2 h, >8.7 at 6 h, and >10.4 at 12 h after the commencement of HFNC therapy were all meaningful predictors of HFNC success in extubated patients. In addition, we found that an integrated model including sex, BMI, total duration of ventilator care, and the ROX index could be applied in real-world practice to predict HFNC success within 72 h after extubation.

Almost 20% of extubated patients require reintubation, and the mortality rate is high among those who are reintubated [11]. To prevent reintubation, HFNC therapy can be considered immediately after extubation. Previous studies have shown that the HFNC is a more useful tool to prevent hypoxemia after extubation in patients at low risk (and even some at high risk) for reintubation than conventional oxygen therapy [1,2,4,8,12,15]. However, HFNC therapy may delay reintubation in patients with aggravated clinical conditions, and this could be associated with an increased mortality rate [13,14,15]. In addition, there are no appropriate tools to evaluate the success or failure of HFNC therapy after planned extubation in real clinical practice. In previous studies, the ROX index was evaluated and validated as a useful, simple tool to predict HFNC success; however, the study populations in these previous studies were limited to patients with pneumonia-related ARF [16,17]. This study demonstrated that the ROX index could be used to predict HFNC success in all extubated patients with various diseases as well as pneumonia-related ARF. In addition, most patients in this study were elderly and had many comorbidities, including neurological diseases. Considering that the proportion of elderly patients with various comorbidities in ICUs is rapidly increasing, the characteristics of the patients in this study were similar to those encountered in medical ICUs in general [23]. Therefore, our results regarding the ROX index are applicable to all extubated patients under real-world conditions.

In this study, an ROX index of >8.7 at 2 h, >8.7 at 6 h, and >10.4 at 12 h were meaningful predictors of HFNC success within 72 h of extubation. The ROX index cut-off value in this study was higher than in previous reports. As the ROX index consists of the SpO_2_/FiO_2_/respiration rate, this result indicates that reintubation may be considered even in patients with relatively stable conditions. Therefore, other factors may need to be taken into consideration in addition to the ROX index to determine whether to reintubate. The integrated model including the ROX index, sex, BMI, and the total duration of ventilator care had a better predictive capability for HFNC success than the ROX index alone. Low BMI and a long duration of ventilator care are associated with ICU-acquired weakness, which is associated with reintubation after extubation [24,25,26]. Therefore, a high BMI and a short duration of ventilator care before extubation could be associated with HFNC success. In the present study, the AUC of the integrated model including the ROX index, sex, BMI, and the total duration of ventilator care improved to 0.858, indicating that this model could be useful for predicting HFNC success after extubation in actual clinical practice. In the present study, the median time from extubation to reintubation was 17 h. Therefore, intensivists should measure serial ROX indices for the first 12 h after extubation to predict HFNC success or failure and should closely evaluate whether reintubation should be performed in male patients, patients with BMIs ≤ 21.5, patients with ROX indices below the cut-off values, and patients with a total duration of ventilator care of >8 days.

Recently, the number of critically ill patients with coronavirus disease 2019 (COVID-19) has increased. During the COVID-19 pandemic, HFNC therapy has been used more frequently than conventional oxygen therapy to manage ARF because of the maintenance of oxygen concentration despite a patient’s declining status and its easier manipulation [27,28,29,30,31]. For this reason, clinicians may also prefer to apply HFNC in extubated patients. In such clinical settings, our model could be important in terms of applicability for all extubated patients with the commencement of HFNC therapy and in terms of the reduction of concerns regarding infection when this model is applied to evaluate whether reintubation should be performed because it consisted of EMR-based parameters (i.e., ROX index, sex, BMI, and the total duration of ventilator care). Further studies are needed to evaluate the effectiveness of this model for predicting HFNC success in extubated COVID-19 patients.

This study has several limitations. First, the study has several biases due to its retrospective observational cohort study design. To reduce selection bias, all patients who started HFNC therapy after extubation were included in this study, and research nurses and statisticians other than the authors participated in data collection from EMRs and statistical analysis. To reduce information bias, well-trained research nurses collected our data using standardized protocols for data collection, and patients with missing data above 20% of each variable were excluded from the analysis. Nevertheless, further studies are needed to validate our results because this study could still have been subject to several biases due to its retrospective design. Second, the number of patients in this study was relatively small. However, the number was statistically sufficient to evaluate the clinical value of the ROX index and to develop an integrated model including the ROX index for predicting HFNC success within 72 h after planned extubation. Third, the reintubation rate in this study was higher than in previous studies because our study population included many patients at high risk for reintubation (e.g., older age and patients with many comorbid diseases). According to the international guidelines, patients at high risk for reintubation should receive NIV after extubation. However, many clinicians prefer to use HFNC therapy in these patients because of a number of concerns, such as poor mask fitting, claustrophobia, patient/ventilator asynchrony, and intolerance [10,32,33,34,35,36]. Therefore, as our study population included elderly patients with various comorbidities, it may be useful for practical application. Finally, this study did not have a validation dataset. Therefore, our results may not be generalizable but may help to determine whether a patient treated with HFNC therapy after extubation requires reintubation. Despite several limitations, an integrated model including the ROX index could be a useful, simple bedside tool for predicting HFNC success after extubation. In addition, our results are applicable to all extubated patients under real-world conditions because the characteristics of the patients in this study were similar to those encountered in the medical ICU.

In conclusion, the ROX index at 2, 6, and 12 h could be applied to all extubated patients to predict HFNC success after planned extubation. To improve its predictive power, we should also consider an integrated model consisting of the ROX index, sex, BMI, and the total duration of ventilator care. Additional prospective studies with larger cohorts are warranted to validate our findings.

## Figures and Tables

**Figure 1 jcm-10-03513-f001:**
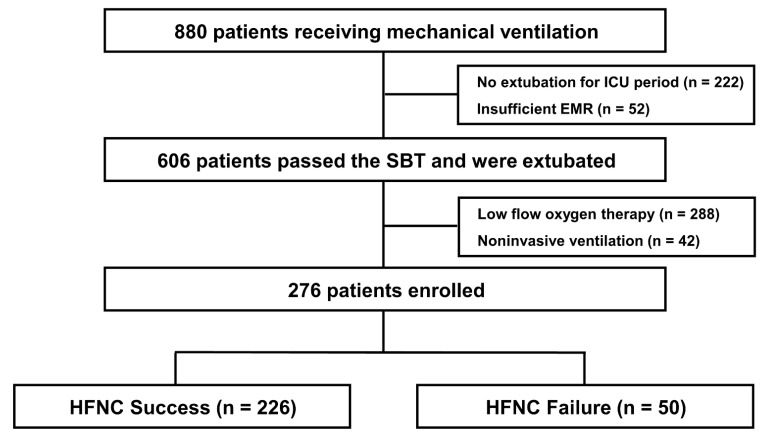
Flow chart of enrolled patients. ICU, intensive care unit; EMR, electric medical records; SBT, spontaneous breathing trial; HFNC, high-flow nasal cannula.

**Figure 2 jcm-10-03513-f002:**
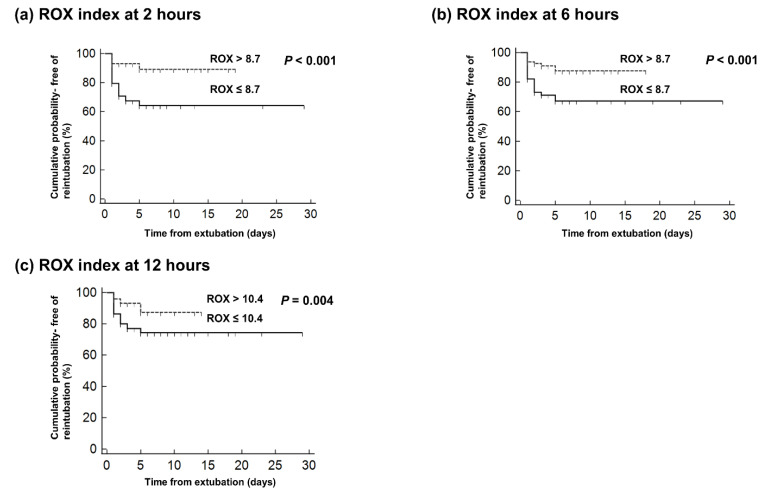
Kaplan–Meier plots showing the cumulative probability of remaining free of reintubation according to the cut-off ROX index value at (**a**) 2 h, (**b**) 6 h, and (**c**) 12 h after the commencement of HFNC therapy in extubated patients.

**Figure 3 jcm-10-03513-f003:**
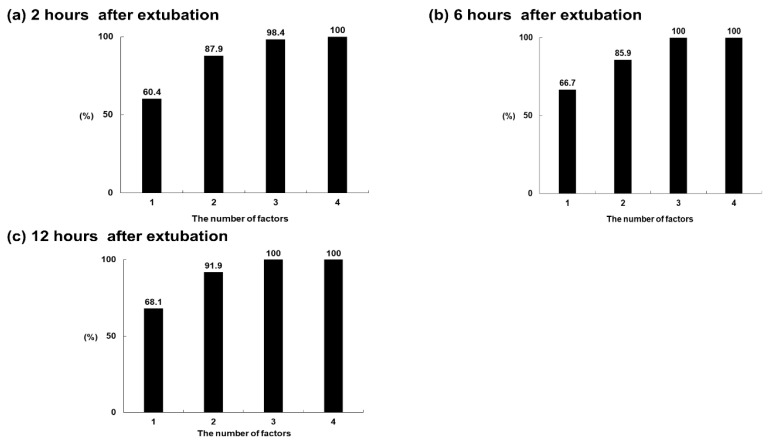
The changes in predictive value for HFNC success within 72 h according to the number of factors associated with HFNC success at each time point. (**a**) Changes in predicted values according to changes in factors, such as an ROX index at 2 h of >8.7, female sex, BMI >21.53 kg/m^2^, and a total duration of ventilator management of ≤8 days. (**b**) Changes in predicted values according to changes in factors, such as an ROX index at 6 h of >8.7, female sex, BMI >21.53 kg/m^2^, and a total duration of ventilator management of ≤8 days. (**c**) Changes in predicted values according to changes in factors, such as an ROX index at 12 h of >8.7, female sex, BMI > 21.53 kg/m^2^, and a total duration of ventilator management of ≤8 days. Sequential increases in factors associated with the success of HFNC (e.g., ROX index at 2 h, 6 h, and 12 h, female sex, BMI > 21.53 kg/m^2^, and a total duration of ventilator care ≤8 days) improved the predictive capability for HFNC success within 72 h.

**Table 1 jcm-10-03513-t001:** Clinical characteristics of patients in this study.

Variables	HFNC Outcome within 72 H		*p* Value
	Success (*n* = 226)	Failure (*n* = 50)	
Age *	77 (67–84)	78 (71–85)	0.216
Male sex	136 (60.2)	40 (80)	0.009
Body mass index *	22 (19–24)	20 (17–22)	0.002
APACHE II score at admission *	30 (27–34)	30 (28–33)	0.626
Charlson comorbidity index *	6 (4–8)	6 (4–8)	0.989
Comorbidities			
Dementia	28 (12.4)	6 (12)	1.000
Stroke	36 (15.9)	6 (12)	0.663
Parkinson’s disease	12 (5.3)	2 (4)	1.000
Seizure disorder	4 (1.8)	2 (4)	0.298
Diabetes mellitus	83 (36.7)	13 (26)	0.189
Chronic kidney disease	52 (23)	9 (18)	0.572
Solid cancer	64 (28.3)	13 (26)	0.862
Hematologic malignancy	21 (9.3)	4 (8)	1.000
Cardiovascular disease	65 (28.8)	12 (24)	0.602
COPD	39 (17.3)	11 (22)	0.422
Diagnosis at admission			
Cardiovascular disease	11 (4.9)	1 (2.0)	0.700
Pulmonary disease	146 (64.6)	31 (62)	0.746
Gastrointestinal disease	4 (1.8)	0 (0)	1.000
Neurologic disease	14 (6.2)	8 (16)	0.038
Renal disease	2 (0.9)	0 (0)	1.000
Other disease	49 (21.7)	10 (20)	0.851
Immunosuppressive therapy	13 (5.8)	6 (12)	0.125
Clinical status at the time of extubation			
Systolic blood pressure (mmHg) *	128 (112–143)	127 (111–146)	0.895
Diastolic blood pressure (mmHg) *	70 (60–81)	72 (62–81)	0.986
Heart rate (beats/min) *	93 (80–104)	96 (83–111)	0.098
Respiratory rate (breaths/min) *	21 (17–25)	23 (17–29)	0.055
Pao_2_/Fio_2_ *	310 (239–404)	264 (185–356)	0.015
SOFA score *	6 (4–8)	7 (5–9)	0.052
GCS score *	14 (12–14)	13 (12–14)	0.353
Analgesia use †	96 (42.5)	27 (54)	0.158
Vasopressor use ‡	59 (26.1)	19 (38)	0.117
Hemodialysis use §	22 (9.7)	7 (14)	0.443
Total duration of ventilator care *	5 (3–9)	12 (6–16)	<0.001
Total duration of HFNC use, hours *	42 (21–86)	17 (5–25)	<0.001

Abbreviations: HFNC, high-flow nasal cannula; APACHE II, acute physiology and chronic health evaluation II; COPD, chronic obstructive pulmonary disease; Pao_2_, partial pressure of oxygen; Fio_2_, fraction of inspired oxygen; SOFA, sequential organ failure assessment; GCS, Glasgow Come Scale. * Data are presented as medians (25th percentile–75th percentiles). Other variables are presented as numbers (percentages). † Low-dose remifentanil was used as an analgesic. ‡ Low-dose norepinephrine or low-dose dopamine was used as a vasopressor. § Conventional hemodialysis or continuous renal replacement therapy was used as hemodialysis.

**Table 2 jcm-10-03513-t002:** Diagnostic accuracies of different respiratory variables at different time points for predicting the success of high-flow nasal cannula therapy within 72 h.

Variables	Time (Hours)	AUROC	95% CI	*p* Value
Spo_2_/Fio_2_	Before extubation	0.622	0.562–0.680	0.004
	2	0.643	0.583–0.701	0.001
	6	0.619	0.555–0.681	0.022
	12	0.624	0.559–0.686	0.014
RR, breaths/min	Before extubation	0.595	0.535–0.654	0.042
	2	0.625	0.564–0.683	0.002
	6	0.708	0.646–0.764	<0.001
	12	0.678	0.614–0.738	<0.001
Paco_2_, mmHg	Before extubation	0.504	0.443–0.564	0.934
	2	0.512	0.450–0.573	0.814
	6	0.568	0.487–0.647	0.312
	12	0.506	0.431–0.582	0.923
Flow, L/min	Before extubation			
	2	0.601	0.540–0.660	0.003
	6	0.600	0.536–0.662	0.014
	12	0.584	0.518–0.648	0.064
Sp_O2_, %	Before extubation	0.546	0.485–0.606	0.288
	2	0.553	0.492–0.614	0.248
	6	0.554	0.489–0.617	0.293
	12	0.582	0.516–0.646	0.109
Fio_2_	Before extubation	0.622	0.562–0.680	0.003
	2	0.631	0.570–0.688	0.002
	6	0.611	0.547–0.673	0.026
	12	0.608	0.542–0.671	0.029
Lactate, mmol/L	Before extubation			
	2	0.561	0.486–0.634	0.363
	6	0.542	0.443–0.639	0.566
	12	0.632	0.541–0.715	0.094
ROX index	Before extubation			
	2	0.709	0.651–0.763	<0.001
	6	0.707	0.645–0.763	<0.001
	12	0.729	0.668–0.785	<0.001

Abbreviations: AUROC, area under the receiver operating characteristic; CI, confidence interval; Spo_2_, saturation of percutane ous oxygen; Fio_2_, fraction of inspired oxygen; HFNC, high-flow nasal cannula; RR, respiratory rate; Paco_2_, partial pressure of carbon dioxide.

**Table 3 jcm-10-03513-t003:** Cox proportional hazard model analyzing the ROX index at different time points of high-flow nasal cannula therapy application adjusted by potential covariates for the risk for reintubation.

Variables	Hazard Ratio	95% CI	*p* Value
Unadjusted ROX index			
2 h (ROX index > 8.7)	0.27	0.141–0.527	<0.001
6 h (ROX index > 8.7)	0.33	0.167–0.645	0.001
12 h (ROX index > 10.4)	0.37	0.164–0.818	0.014
Adjusted by male			
2 h (ROX index > 8.7)	0.29	0.151–0.566	<0.001
6 h (ROX index > 8.7)	0.34	0.173–0.671	0.002
12 h (ROX index > 10.4)	0.37	0.164–0.818	0.014
Adjusted by BMI			
2 h (ROX index > 8.7)	0.25	0.129–0.485	<0.001
6 h (ROX index > 8.7)	0.30	0.152–0.590	<0.001
12 h (ROX index > 10.4)	0.32	0.144–0.728	0.006
Adjusted by Pao_2_/Fio_2_ before HFNC			
2 h (ROX index > 8.7)	0.27	0.141–0.527	<0.001
6 h (ROX index > 8.7)	0.33	0.167–0.645	0.001
12 h (ROX index > 10.4)	0.37	0.164–0.818	0.014
Adjusted by total duration of ventilator care			
2 h (ROX index > 8.7)	0.30	0.154–0.577	0.001
6 h (ROX index > 8.7)	0.39	0.196–0.778	0.007
12 h (ROX index > 10.4)	0.42	0.187–0.948	0.037

Abbreviations: HFNC, high-flow nasal cannula; CI, confidence interval; BMI, body mass index; Pao_2_, partial pressure of oxygen; Fio_2_, fraction of inspired oxygen.

**Table 4 jcm-10-03513-t004:** Prognostic capabilities of models using the ROX index and potential factors for predicting HFNC success.

ROX Index	Models	C-Index	95% CI	*p* Value
	1	0.709	0.651–0.763	Reference
ROX index at 2 h	2	0.786	0.705–0.867	0.013
Model	3	0.798	0.728–0.868	0.005
	4	0.811	0.745–0.877	0.005
	5	0.707	0.645–0.763	Reference
ROX index at 6 h	6	0.792	0.704–0.880	0.009
Model	7	0.810	0.738–0.883	0.003
	8	0.841	0.777–0.903	0.002
	9	0.729	0.668–0.785	Reference
ROX index at 12 h	10	0.816	0.731–0.901	0.015
Model	11	0.833	0.758–0.907	0.003
	12	0.858	0.789–0.927	0.002

Models 1–4 included the ROX index at 2 h and other factors, models 5–8 included the ROX index at 6 h and other factors, and models 9–12 included the ROX index at 12 h and other factors. Model 1 only included the ROX index at 2 h after HFNC commencement. Model 2 included the ROX index at 2 h and the total duration of ventilator care. Model 3 included the ROX index at 2 h, the total duration of ventilator care, and female sex. Model 4 included the ROX index at 2 h, the total duration of ventilator care, female sex, and body mass index. Model 5 only included the ROX index at 6 h after HFNC commencement. Model 6 included the ROX index at 6 h and the total duration of ventilator care. Model 7 included the ROX index at 6 h, the total duration of ventilator care, and female sex. Model 8 included the ROX index at 6 h, the total duration of ventilator care, female sex, and body mass index. Model 9 only included the ROX index at 12 h after HFNC commencement. Model 10 included the ROX index at 12 h and the total duration of ventilator care. Model 11 included the ROX index at 12 h, the total duration of ventilator care, and female sex. Model 12 included the ROX index at 12 h, the total duration of ventilator care, female sex, and body mass index.

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
