# Peer review of "An Integrated Model including the ROX Index to Predict the Success of High-Flow Nasal Cannula Use after Planned Extubation: A Retrospective Observational Cohort Study"

_jcm, 2021, doi:10.3390/jcm10163513_

Round 1
Reviewer 1 Report
Summary of this paper
I have reviewed the manuscript entitled "An integrated model including the ROX index to predict the success of high flow nasal cannula use after planned extubation" by Seok Lee et al. In this retrospective, single-center observational study involving 276 patients requiring High flow nasal cannula (HFNC) after extubation, the authors sought to determine the usefulness of ROX index (SpO2/FiO2/respiratory rate) for predicting HFNC success after extubation. They found that ROX index >8.7 at 2 hours, 17 >8.7 at 6 hours and >10.4 at 12 hours after HFNC commencement was a meaningful predictor of HFNC success within 72 hours of extubation. They also found that an integrated model, including ROX index, sex, body mass index, and total duration of ventilator care, was able to predict HFNC success better than was the model using the ROX index alone. From these observations, they claimed the diagnostic value of the ROX index in these population. First, I admire the author's effort and time spent on this project. However, this manuscript contains several flaws that should be amended. My concerns are listed below.
General comment
1
In several places, this manuscript does not conform the relevant guideline such as STROBE (https://www.strobe-statement.org/index.php?id=strobe-home) and TRIPOD (https://www.equator-network.org/reporting-guidelines/tripod-statement/). Your manuscript would benefit from these guidelines. Please check throughout your manuscript.
Title
2
Indicate the study’s design with a commonly used term in the title.
Introduction
3
State prespecified hypotheses in the introduction section.
Methods
4
Study setting
Please describe each study settings more in details (e.g. closed or semi-closed or open ICU, number of annual ICU admission, number of annual mechanical ventilation, number of annual ambulances, number of hospital beds and ICU beds, number of intensivists and nurses) where the data were collected. I believe such information would help readers to depict the context of your study more accurately.
5
Who planned this study, who enrolled participants, who classified patients into HFNC success and failure, who measured ROX index, and who conducted the statistical analysis? I think if the same researchers are involved in study planning, enrolling, outcome measurement, and statistical analysis, there is a theoretical risk of biased assessment. Who did extract these patients, and who constructed the database and how? The characteristics of this investigator should be described in detail (ex. resident physician, board certified ICU physician).
6
Describe any efforts to address potential sources of bias. For example, blinding is one of the attractive methods to reduce above mentioned biased assessment. If done, please provide who was blinded and how.
7
The authors should provide the characteristics of data source used in this study. How did you conduct quality control of your database? Who entered the data and how could you assure the quality and validity of data?
Sample size
8
Sample size estimation was missing. Explain how the study size was arrived at.
9
How did you consider and deatl with do not attempt resuscitation order?
10
How did you determine the cut off value of ROX index? Using Youden Index or some other methods? Please provide the rationale and clarify in detail.
11
Ethical statement should include the relevant date of the approval.
12
The authors excluded the 52 patients with insufficient medical record. Were these values Missing At Random or Missing Not At Random and why did you think so? What is your opinion on this? If these values were Missing Not At Random, how did you deal with this problem?
Results
13
Table 1
Baseline demographic and clinical characteristics of participants.
Several vital information is missing, such as vital signs, conscious level, anesthetics and analgesics, social factors, use of cigarette, APACH II score related parameter (use of catechol amines, kidney dysfunction, etc), liver dysfunction, use of hemodialysis and continuous venovenous hemodialysis, ventilator settings. The reviewer thinks above mentioned clinical information is important confounding factors. Could the authors provide such information?
Discussion
14
Discuss the potential clinical use of the model and implications for future research. Implications for practice, including the intended use and clinical role of the index test.
15
The limitation section needs substantial revision. Please discuss limitations of the study, taking into account sources of potential bias or imprecision. Consider the important limitations and do not just list them but consider their relevance and how they might bias the results. Discuss both direction and magnitude of any potential bias. Please indicate the strength of this paper and future research direction.
16
How about the cost and versatility of the ROX measurement? What is your opinion?
17
The relevancy of your study in the era of the COVID-19 pandemic should be discussed more rigorously.
Author Response
Comment (C) 1. In several places, this manuscript does not conform the relevant guideline such as STROBE (https://www.strobe-statement.org/index.php?id=strobe-home) and TRIPOD (https://www.equator-network.org/reporting-guidelines/tripod-statement/). Your manuscript would benefit from these guidelines. Please check throughout your manuscript.
Response (R) 1. We thank you for your comment. We checked the revised manuscript using the guideline of STROBE. We attached STROBE checklist to submit system when we resubmit the revised manuscript.
C2. (Title) Indicate the study’s design with a commonly used term in the title.
R2. Thank you for your comment. We indicated the study design in the title.
“An integrated model including the ROX index to predict the success of high flow nasal cannula use after planned extubation: A retrospective observational cohort study.” (page 1, line 1-2)
C3. (Introduction) State prespecified hypotheses in the introduction section.
R3. Thank you for your valuable comment. We inserted the following sentences in introduction section.
“As the ROX index has been validated only in patients with pneumonia-related ARF, it cannot be applied to other clinical settings. As the ROX index is an effective bedside tool for predicting HFNC success, it should be validated for use in various clinical settings. We hypothesized that the ROX index would be useful to predict the success of HFNC in extubated patients, and that a model composed of the ROX index and other factors would improve the predictive power on HFNC success in all extubated patients.” (page 3, line 48-53)
C4. (Methods) Study setting
Please describe each study settings more in details (e.g., closed or semi-closed or open ICU, number of annual ICU admission, number of annual mechanical ventilation, number of annual ambulances, number of hospital beds and ICU beds, number of intensivists and nurses) where the data were collected. I believe such information would help readers to depict the context of your study more accurately.
R4. Thank you for your valuable comment. As the reviewer’s comments, we inserted the data of hospital in methods section as the following sentences.
“This was a retrospective observational cohort study of patients who were immediately treated with HFNC after planned extubation in the medical intensive care unit (ICU) at Korea Medical Center, a 1075-bed tertiary referral hospital in Seoul, Republic of Korea, between January 2017 and December 2019. This hospital has a 20-bed semi-closed medical ICU with an annual ICU admission rate of 400 patients and an annual ventilator application rate of 300 patients. The ratio of patients to nurses is approximately 2.5:1 and three intensivists administer nearly all patients in the medical ICU. All patients in this study were progressively weaned from mechanical ventilation at the discretion of the intensivist based on the weaning protocol of the hospital.” (page 5, line 60-68)
C5. (Methods) Who planned this study, who enrolled participants, who classified patients into HFNC success and failure, who measured ROX index, and who conducted the statistical analysis? I think if the same researchers are involved in study planning, enrolling, outcome measurement, and statistical analysis, there is a theoretical risk of biased assessment. Who did extract these patients, and who constructed the database and how? The characteristics of this investigator should be described in detail (ex. resident physician, board certified ICU physician).
C6. (Methods) Describe any efforts to address potential sources of bias. For example, blinding is one of the attractive methods to reduce above mentioned biased assessment. If done, please provide who was blinded and how.
R5, 6. We understand reviewer’s concerns. Young Seok Lee and Je Hyeong Kim planned this study. Sung Won Chang and research nurses collected patient data. Collected data were statistically analyzed by Young Seok Lee and a statistician (Prof. Soon Young Hwang) at the Korean Medical Center. All authors analyzed and interpreted the results of this study and approved the final version of the manuscript. In addition, well-trained research nurses collected our data based on standardized protocols for data collection and patients with missing data above 20% of each variable were excluded from the analysis. Nevertheless, this study had several biases due to its retrospective observational cohort study design. We inserted the following sentences for biases in limitation section.
“First, the study had several biases due to its retrospective observational cohort study design. To reduce selection bias, all patients who started HFNC after extubation were included in this study, and research nurses and statisticians other than the authors participated in data collection based on EMR and statistical analysis. To reduce information bias, well-trained research nurses collected our data based on standardized protocols for data collection and patients with missing data above 20% of each variable were excluded from the analysis. Nevertheless, further studies are needed to validate our results because this study could still have been subject to several biases due to its retrospective design.” (page 24, line 345-352)
C7. (Methods) The authors should provide the characteristics of data source used in this study. How did you conduct quality control of your database? Who entered the data and how could you assure the quality and validity of data?
R7. Thank you for your comment. This data was collected retrospectively through medical chart review. Sung Won Chang and research nurses participated in data collection and Young Seok Lee reviewed the data for quality control and confirmed it. In addition, to reduce information bias, well-trained research nurses collected our data based on standardized protocols for data collection and patients with missing data above 20% of each variable were excluded from the analysis. (e.g., ROX index at 24 h).
C8. (Methods) Sample size estimation was missing. Explain how the study size was arrived at.
R8. This study was a retrospective study. Therefore, sample size was not calculated. Considering the fact that nearly 20% of extubated patients go through reintubation, we believe sample size in this study is sufficient enough to evaluate the results of this study because the reintubation rate was 18%.
C9. (Methods) How did you consider and deal with do not attempt resuscitation order?
R9. Thank you for your valuable comment. This study focused on the clinical value of the ROX index to predict success of HFNC therapy within 72 h of commencement in extubated patients, and the development of an integrated model including the ROX index to improve the predictive power for HFNC success. Therefore, do-not-resuscitation did not affect the outcome because HFNC-failed patients were reintubated. In our hospital, we applied a simple mask (e.g., Respiflo) to patients who wrote written do-not-intubate paper before extubation for the effective use of medical resources.
C10. (Methods) How did you determine the cut off value of ROX index? Using Youden Index or some other methods? Please provide the rationale and clarify in detail.
R10. Thank you for your comment. As reviewer’s comment, we inserted the following sentences in statistically section.
“The cut off levels of the ROX index and other factors in the integrated model were calculated using the Youden index of ROC analysis to determine the optimal diagnostic accuracy for these factors.” (page 7, line 123-125; page 8, line 126)
C11. (Methods) Ethical statement should include the relevant date of the approval.
R11. Thank you for your comment. As reviewer’s comment, we inserted the approval date in ethical statement.
“This study was approved by the Institutional Review Board of Korea Medical Center (approval no.: 2020GR0518, November 4, 2020).” (page 5, line 72-73)
C12. (Methods) The authors excluded the 52 patients with insufficient medical record. Were these values Missing at Random or Missing Not at Random and why did you think so? What is your opinion on this? If these values were Missing Not at Random, how did you deal with this problem?
R12. We excluded the 52 patients with insufficient medical record because these patients didn’t have serial data to calculate ROX index. Patients without serial data may be in a less severe clinical state after extubation than those with serial data because these patients were not checked vital sign regularly. This is the limitation of retrospective cohort study design.
C13. (Results) Table 1. Baseline demographic and clinical characteristics of participants.
Several vital information is missing, such as vital signs, conscious level, anesthetics and analgesics, social factors, use of cigarette, APACH II score related parameter (use of catechol amines, kidney dysfunction, etc), liver dysfunction, use of hemodialysis and continuous venovenous hemodialysis, ventilator settings. The reviewer thinks above mentioned clinical information is important confounding factors. Could the authors provide such information?
R13. Thank you for your comment. As reviewer’s comment, we provided these data in table 1 and table S4. These variable in our study did not affect our results because of statistically insignificant.
Table 1. Clinical characteristics of patients in this study
|
Variables |
HFNC Outcome within 72 hours |
p value |
|
|
|
Success (N = 226) |
Failure (N = 50) |
|
|
Age* |
77 (67-84) |
78 (71-85) |
0.216 |
|
Male sex |
136 (60.2) |
40 (80) |
0.009 |
|
Body mass index* |
22 (19-24) |
20 (17-22) |
0.002 |
|
APACHE II score at admission* |
30 (27-34) |
30 (28-33) |
0.626 |
|
Charlson comorbidity index* |
6 (4-8) |
6 (4-8) |
0.989 |
|
Comorbidities |
|
|
|
|
Dementia |
28 (12.4) |
6 (12) |
1.000 |
|
Stroke |
36 (15.9) |
6 (12) |
0.663 |
|
Parkinson disease |
12 (5.3) |
2 (4) |
1.000 |
|
Seizure disorder |
4 (1.8) |
2 (4) |
0.298 |
|
Diabetes mellitus |
83 (36.7) |
13 (26) |
0.189 |
|
Chronic kidney disease |
52 (23) |
9 (18) |
0.572 |
|
Solid cancer |
64 (28.3) |
13 (26) |
0.862 |
|
Hematologic malignancy |
21 (9.3) |
4 (8) |
1.000 |
|
Cardiovascular disease |
65 (28.8) |
12 (24) |
0.602 |
|
COPD |
39 (17.3) |
11 (22) |
0.422 |
|
Diagnosis at admission |
|
|
|
|
Cardiovascular disease |
11 (4.9) |
1 (2.0) |
0.700 |
|
Pulmonary disease |
146 (64.6) |
31 (62) |
0.746 |
|
Gastrointestinal disease |
4 (1.8) |
0 (0) |
1.000 |
|
Neurologic disease |
14 (6.2) |
8 (16) |
0.038 |
|
Renal disease |
2 (0.9) |
0 (0) |
1.000 |
|
Other disease |
49 (21.7) |
10 (20) |
0.851 |
|
Immunosuppressive therapy |
13 (5.8) |
6 (12) |
0.125 |
|
Clinical status at the time of extubation |
|
|
|
|
Systolic blood pressure (mmHg) * |
128 (112-143) |
127 (111-146) |
0.895 |
|
Diastolic blood pressure (mmHg) * |
70 (60-81) |
72 (62-81) |
0.986 |
|
Heart rate (beats/min) * |
93 (80-104) |
96 (83-111) |
0.098 |
|
Respiratory rate (breaths/min) * |
21 (17-25) |
23 (17-29) |
0.055 |
|
PaO2/FiO2* |
310 (239-404) |
264 (185-356) |
0.015 |
|
SOFA score* |
6 (4-8) |
7 (5-9) |
0.052 |
|
GCS score* |
14 (12-14) |
13 (12-14) |
0.353 |
|
Analgesia use† |
96 (42.5) |
27 (54) |
0.158 |
|
Vasopressor use‡ |
59 (26.1) |
19 (38) |
0.117 |
|
Hemodialysis use§ |
22 (9.7) |
7 (14) |
0.443 |
|
Total duration of ventilator care* |
5 (3-9) |
12 (6-16) |
<0.001 |
|
Total duration of HFNC use, hours* |
42 (21-86) |
17 (5-25) |
<0.001 |
Abbreviations: HFNC, high flow nasal cannula; APACHE II, acute physiology and chronic health evaluation II; COPD, chronic obstructive pulmonary disease; PaO2, partial pressure of oxygen; FiO2, fraction of inspired oxygen; SOFA, sequential organ failure assessment; GCS, Glasgow Come Scale.
* Data are presented as median (25th percentile-75th percentile). Other variables are presented as number (percent).
† Low-dose remifentanil was used as an analgesia.
‡ Low-dose norepinephrine or low-dose dopamine was used as a vasopressor.
§Conventional hemodialysis or continuous renal replacement therapy was used as hemodialysis.
Table S4. Factors for predicting reintubation within 72 hours after HFNC commencement using Cox proportional hazards model
|
Variables |
Hazard Ratio |
95% CI |
P value |
|
Age |
1.01 |
0.985-1.033 |
0.474 |
|
Male sex |
2.38 |
1.188-4.754 |
0.014 |
|
Body mass index |
0.90 |
0.832-0.965 |
0.004 |
|
APACHE II score at admission |
1.00 |
0.956-1.044 |
0.961 |
|
SOFA score, at the start of HFNC |
1.08 |
0.969-1.192 |
0.172 |
|
Charlson comorbidity index |
1.00 |
0.908-1.094 |
0.945 |
|
Comorbidities |
|
|
|
|
Dementia |
0.93 |
0.395-2.179 |
0.864 |
|
Stroke |
0.65 |
0.278-1.535 |
0.329 |
|
Parkinson disease |
0.73 |
0.178-3.018 |
0.667 |
|
Seizure disorder |
2.38 |
0.577-9.780 |
0.231 |
|
Diabetes mellitus |
0.60 |
0.318-1.128 |
0.112 |
|
Chronic kidney disease |
0.70 |
0.341-1.450 |
0.340 |
|
Solid cancer |
0.85 |
0.451-1.598 |
0.612 |
|
Hematologic malignancy |
0.83 |
0.297-2.297 |
0.715 |
|
Cardiovascular disease |
0.87 |
0.455-1.668 |
0.678 |
|
COPD |
1.22 |
0.622-2.374 |
0.569 |
|
Immunosuppressive therapy |
2.02 |
0.862-4.747 |
0.106 |
|
Clinical status at the time of extubation |
|
|
|
|
Systolic blood pressure (mmHg) |
1.00 |
0.989-1.014 |
0.865 |
|
Diastolic blood pressure (mmHg) |
0.99 |
0.982-1.016 |
0.898 |
|
Heart rate (beats/min) |
1.01 |
0.999-1.026 |
0.060 |
|
Respiratory rate (breaths/min) |
1.03 |
0.985-1.067 |
0.219 |
|
PaO2/FiO2 |
0.99 |
0.994-1.000 |
0.027 |
|
SOFA score |
1.08 |
0.969-1.192 |
0.172 |
|
GCS score |
0.98 |
0.877-1.096 |
0.726 |
|
Analgesia use |
1.45 |
0.828-2.524 |
0.195 |
|
Vasopressor use |
1.58 |
0.892-2.796 |
0.117 |
|
Hemodialysis use |
1.34 |
0.601-2.975 |
0.476 |
|
Total duration of ventilator care |
1.08 |
1.044-1.109 |
<0.001 |
Abbreviations: HFNC, high flow nasal cannula; CI, confidence interval; APACHE II, acute physiology and chronic health evaluation II; SOFA, sequential organ failure assessment; COPD, chronic obstructive pulmonary disease; PaO2, partial pressure of oxygen; FiO2, fraction of inspired oxygen; SOFA, sequential organ failure assessment; GCS, Glasgow Come Scale
C14. (Discussion) Discuss the potential clinical use of the model and implications for future research. Implications for practice, including the intended use and clinical role of the index test.
R14. Thank you for your valuable comment. As reviewer’s comment, we inserted the following sentences in discussion section.
“This study demonstrated that the ROX index could be used to predict HFNC success in all extubated patients with various diseases as well as pneumonia-related ARF. In addition, most of the patients in this study were elderly and had many comorbidities, including neurological diseases. Considering that the proportion of elderly patients with various comorbidities in the ICU is rapidly increasing, the characteristics of the patients in this study were similar to those encountered in the medical ICU [23]. Therefore, our results regarding the ROX index are applicable to all extubated patients under real-world conditions.” (page 22, line 307-311; page 23, line 312-314)
“The ROX index cut off value in this study was higher than in previous reports. As the ROX index consists of the SpO2/FiO2/respiration rate, this result indicates that reintubation may be considered even in patients with relatively stable condition. Therefore, other factors may need to be taken into consideration in addition to the ROX index to determine whether to reintubate. The integrated model including the ROX index, sex, BMI, and the total duration of ventilator care had better predictive capability of HFNC success than the ROX index alone.” (page 23, line 316-322)
“In the present study, the AUC of the integrated model including the ROX index, sex, BMI, and total duration of ventilator care improved to 0.858, indicating that this model could be useful for predicting HFNC success after extubation in actual clinical practice. In the present study, the median time from extubation to reintubation was 17 h. Therefore, intensivists should measure serial ROX indices for the first 12 h after extubation to predict HFNC success or failure and should closely evaluate whether reintubation should be performed in male patients, patients with BMI ≤ 21.5, patients with ROX indices below the cutoff values, and patients with total duration of ventilator care > 8 days.” (page 23, line 325-333)
C15. (Discussion) The limitation section needs substantial revision. Please discuss limitations of the study, taking into account sources of potential bias or imprecision. Consider the important limitations and do not just list them but consider their relevance and how they might bias the results. Discuss both direction and magnitude of any potential bias. Please indicate the strength of this paper and future research direction.
R15. Thank you for your valuable comment. As reviewer’s comment, we inserted the following sentences in limitation section.
“This study had several limitations. First, the study had several biases due to its retrospective observational cohort study design. To reduce selection bias, all patients who started HFNC after extubation were included in this study, and research nurses and statisticians other than the authors participated in data collection based on EMR and statistical analysis. To reduce information bias, well-trained research nurses collected our data based on standardized protocols for data collection and patients with missing data above 20% of each variable were excluded from the analysis. Nevertheless, further studies are needed to validate our results because this study could still have been subject to several biases due to its retrospective design. Second, the number of patients in this study was relatively small. However, the number was statistically sufficient to evaluate the clinical value of the ROX index and to develop an integrated model including the ROX index for predicting HFNC success within 72 h after planned extubation. Third, the reintubation rate in this study was higher than in previous studies because our study population included many patients at high risk for reintubation (e.g., older age and patients with many comorbid diseases). According to the international guidelines, patients at high risk for reintubation should receive NIV after extubation. However, many clinicians prefer to use HFNC in these patients due to a number of concerns, such as poor mask-fitting, claustrophobia, patient/ventilator asynchrony, and intolerance [10,32-36]. Therefore, as our study population included elderly patients with various comorbidities, it may be useful for practical application. Finally, this study did not have a validation dataset. Therefore, our results may not be generalizable, but may help to determine whether a patient treated with HFNC after extubation requires reintubation. Despite several limitations, an integrated model including the ROX index could be a useful, simple bedside tool for predicting HFNC success after extubation. In addition, our results are applicable to all extubated patients under real-world conditions because the characteristics of the patients in this study were similar to those encountered in the medical ICU.” (page 24, line 345-357; page 25, line 358-369)
C16. (Discussion) How about the cost and versatility of the ROX measurement? What is your opinion?
R16. Considering that the ROX index is an effective bedside tool for predicting HFNC success using relatively simple parameters, we think that the ROX index is a cost-effective tool. As the ROX index has been validated only in patients with pneumonia-related ARF, it cannot be applied to other clinical settings. As the ROX index is cost-effective bedside tool for predicting HFNC success, we think that it should be validated for use in various clinical settings.
C17. (Discussion) The relevancy of your study in the era of the COVID-19 pandemic should be discussed more rigorously.
R17. Thank you for your valuable comment. As reviewer’s comment, we inserted the following sentences in discussion section.
“Recently, the number of critically ill patients with coronavirus disease 2019 (COVID-19) has increased. During the COVID-19 pandemic, HFNC has been used more frequently than conventional oxygen therapy to manage ARF because of the maintenance of oxygen concentration despite a patient’s declining status and its easier manipulation. For this reason, clinicians may also prefer to apply HFNC in extubated patients. In such clinical settings, our model could be important in terms of applicability for all extubated patients with commencement of HFNC therapy, and in terms of the reduction of concerns regarding infection when this model is applied to evaluate whether reintubation should be performed because it consisted of EMR-based parameters (i.e., ROX index, sex, BMI, and total duration of ventilator care). Further studies are needed to evaluate the effectiveness of this model for predicting HFNC success in extubated COVID-19 patients.” (page 23, line 334; page 24, line 335-344)

Reviewer 2 Report
The originality of the article is important. Saving human lives is very important.
The value of the intervention is greatly reduced to a very small group of patients. Of all the possible cases only 1/4 of them are using the technique.
With all the data in Table 1, a statistical study could be made of the cases that the HFNC is useful and be able to establish a correlation not only with age, and BMI and would give a greater value to the study.
Figure 2. The interpretation is complex. it requires further explanation.
The conclusions better define when the ROX index should be applied and in what type of patient the results will be more optimal.
Author Response
Comment. The originality of the article is important. Saving human lives is very important.
The value of the intervention is greatly reduced to a very small group of patients. Of all the possible cases only 1/4 of them are using the technique.
With all the data in Table 1, a statistical study could be made of the cases that the HFNC is useful and be able to establish a correlation not only with age, and BMI and would give a greater value to the study.
Figure 2. The interpretation is complex. it requires further explanation.
The conclusions better define when the ROX index should be applied and in what type of patient the results will be more optimal.
Response.
Thank you for your valuable comment. The number of patients in this study was relatively small. However, the number was statistically sufficient to evaluate the clinical value of the ROX index and to develop an integrated model including the ROX index for predicting HFNC success within 72 h after planned extubation. In addition, the study had several biases due to its retrospective observational cohort study design. To reduce selection bias, all patients who started HFNC after extubation were included in this study, and research nurses and statisticians other than the authors participated in data collection based on EMR and statistical analysis. To reduce information bias, well-trained research nurses collected our data based on standardized protocols for data collection and patients with missing data above 20% of each variable were excluded from the analysis. Nevertheless, further studies are needed to validate our results because this study could still have been subject to several biases due to its retrospective design. Despite several limitations, an integrated model including the ROX index could be a useful, simple bedside tool for predicting HFNC success after extubation. In addition, our results are applicable to all extubated patients under real-world conditions because the characteristics of the patients in this study were similar to those encountered in the medical ICU. We inserted these sentences in limitation section. In addition, as the reviewer’s comment, we inserted the following sentences in result section.
“Figure 2. Kaplan-Meier plot showing the cumulative probability of remaining free of reintubation according to the cut off ROX index value at (a) 2 hours, (b) 6 hours, and (c) 12 hours after commencement of HFNC therapy in extubated patients” (page 16, line 215-217)
“Figure 3 shows the changes in predictive value of HFNC success within 72 h according to the number of factors associated with HFNC success at each time point. Figure 3(a) shows the changes in predicted values according to changes in factors, such as ROX index at 2 h > 8.7, female sex, BMI > 21.53 kg/m2, and total duration of ventilator management ≤ 8 days. Figure 3(b) shows the changes in predicted values according to changes in factors, such as ROX index at 6 h > 8.7, female sex, BMI > 21.53 kg/m2, and total duration of ventilator management ≤ 8 days. Figure 3(c) shows the changes in predicted values according to changes in factors, such as ROX index at 12 h > 8.7, female sex, BMI > 21.53 kg/m2, and total duration of ventilator management ≤ 8 days. Sequential increases in factors associated with the success of HFNC (e.g., ROX index at 2 h, 6 h, and 12 h, female sex, BMI > 21.53 kg/m2, total duration of ventilator care ≤ 8 days) improved the predictive capability of HFNC success within 72 h. The integrated model including all factors showed good predictive ability reaching almost 100%” (page 19, line 263-264; page 20, line 265-275)
“In conclusion, the ROX index at 2, 6, and 12 h could be applied to all extubated patients to predict HFNC success after planned extubation. To improve its predictive power, we should also consider an integrated model consisting of ROX index, sex, BMI, and total duration of ventilator care. Additional prospective studies with larger cohorts are warranted to validate our findings.” (page 25, line 370-374)

Round 2
Reviewer 1 Report
Thank you for your effort and time spent on this revision. The reviewer thinks these amendments improved the scientific values of this manuscript, as well as readability. My remaining comments are just minor points.
There are spelling inconsistency in FiO2, H2O, SpO2 and some other place.
For example, in page 3 lines 93 to 118, several word such as FiO2, H2O, and SpO2 are displayed not using subscript. In Table 2, the term SpO2 is displayed as Spo2. There are some other similar errors. Please correct.
Author Response
Thank you for your comment. The authors think that your comments improved the scientific values of this manuscript, as well as readability.
As the reviewer's comment, we corrected the inconsistency in FiO2, H2O, SpO2 in revised manuscript. The changes in the revised manuscript are highlighted in red.
Thank you for your valuable comment.